# Effects of Mixed Organic Acids and Essential Oils in Drinking Water on Growth Performance, Intestinal Digestive Capacity, and Immune Status in Broiler Chickens

**DOI:** 10.3390/ani14152160

**Published:** 2024-07-24

**Authors:** Yuanyang Dong, Xulong Gao, Chenqi Qiao, Miaomiao Han, Zhiqiang Miao, Ci Liu, Lei Yan, Jianhui Li

**Affiliations:** 1College of Animal Science, Shanxi Agricultural University, Taigu 030800, China; yuanyangdongemail@126.com (Y.D.); 15968866107@163.com (X.G.); 18835798693@163.com (C.Q.); h_miaomiao2019@163.com (M.H.); mzhq1981@163.com (Z.M.); 2College of Veterinary Medicine, Shanxi Agricultural University, Taigu 030800, China; liuci1988@163.com; 3New Hope Liuhe Co., Ltd., Beijing 100102, China; yanleimy@163.com

**Keywords:** acidified drinking water, essential oil, broiler, immune status, intestinal digestive capacity

## Abstract

**Simple Summary:**

Simple Summary: The ban on the use of antibiotics as growth promoters has compelled the exploration of other alternatives. Acidifiers and essential oils are promising alternatives for antibiotics in poultry production, especially acidifiers provided in drinking water as they are more convenient and reduce the potential corrosion of processing equipment and loss during granulation. This study investigated the effect of acidifier and essential oil supplementation in drinking water on growth performance, intestinal digestive capacity, and immune status in broiler chickens. The results showed that a combination of acidifiers (with formic acid and propionic acid as active ingredients) and essential oils (with eucalyptus essential oil, mint essential oil, and cinnamaldehyde as active ingredients) improved the immune function of broilers and reduced the growth of potential pathogenic microorganisms in the waterline and excreta.

**Abstract:**

In order to evaluate the effects of acidifiers and essential oils in drinking water on growth, intestinal digestive capacity, and immune status in broilers, a total of 480, 1-day-old Arbore Acres broilers were randomly assigned to four treatments including normal tap water (Ctr) and tap water supplemented with acidifier I (ACI), acidifier I and essential oils (ACI+EO), and acidifier II (ACII). Both ACI+EO and ACII increased final body weight. The pH value of the crop and gizzards was reduced by ACI+EO, and ACII decreased the pH values of the proventriculus and gizzards (*p* < 0.05). Compared with control group, ACI, ACI+EO, ACII significantly enhanced lipase activity in jejunum but ACII decreased the level of serum total cholesterol and total triglyceride (*p* < 0.05). Compared with the control group, ACI+EO and ACII significantly increased the relative weight of the spleen, increased the level of serum IgA and IgM, and decreased *E. coli* in excreta, while ACII significantly decreased *Salmonella* in excreta (*p* <0.05). All treatments significantly increased *Lactobacillus* in excreta. In conclusion, ACI+EO improved immune status and ACII was effective in reducing *Salmonella* and promoting *Lactobacillus*, contributing to intestinal health.

## 1. Introduction

The prohibition of antimicrobial growth performance promoters (AGPs) has caused problems in animal performance, including animal performance problems and more subclinical diseases such as dysbacteriosis or feed passage syndrome [1]. Foodborne pathogens such as *Salmonella* commonly exist in poultry production, which may lead to chicken carcass contamination [2]. Thus, it is imperative to find alternatives to antibiotics for the control of potential diseases in livestock production.

Due to the advantages of improving growth performance and health in livestock, mixed organic acids are regarded as promising green alternatives to antibiotics [3]. Organic acids have diverse attributes and functions, such as antimicrobial activity, decreasing the pH of digesta in the gastrointestinal tract, slowing the transit of feed in the intestine, inducing intestinal or pancreatic enzyme secretion and activity, and providing nutrients to the intestine, thus improving gut health and production [4,5]. The most commonly used organic acids in animal nutrition are formic acid, propionic acid, butyric acid, acetic acid, citric acid, malic acid, lactic acid, and benzoic acid, with most of these being weak organic acids and influenced by their pH and pKa value [4]. Acidified drinking water can decrease the pH in the crop, proventriculus, and ileum, increase the height of the jejunum villus, and reduce potentially pathogenic bacteria, thereby improving growth performance [6]. Organic acids can prevent the growth of mold in feed. In particular, short-chain fatty acids are used to control salmonella in poultry [2]. The outer membrane, with the protein channels of Gram-negative bacteria, only allows the diffusion of low-molecular weight compounds into the cytoplasm, making the passage of medium-sized or large molecules more difficult, while Gram-positive bacteria do not possess the same membrane [7]. Thus, formic acid was shown to be active against both *Salmonella* and *E. coli*, and lauric acid, the main active component of coconut fatty acid, was most effective against *C. perfringens*. However, other short-chain fatty acids such as butyric acid have demonstrated indirect activity, inhibiting the expression of virulence factors [7]. Contaminated feed processed with citric acid and lactic acid could reduce the concentration of deoxynivalenol and its derivate 15Ac-DON, which offers a useful tool for reducing mycotoxin load [8]. Citric acid increased metabolizable dry matter and crude protein [9]. Different combinations of organic acids displayed different biological functions and inhibitory effects on microorganisms. For example, formic acid and acetic acid could effectively reduce colonization with *Salmonella* in broilers, and the calcium salt of propionic acid is highly effective against mold [10,11].

Free access to clean drinking water is a primary requirement for optimizing production, especially for meat production because of the limited use of antibiotics [12]. Water systems including drinker lines are vulnerable to microbial contamination even if there is consistent water sanitation, and microbial growths in waterlines were detected throughout the flock grow-out period [12]. Acidified drinking water can prevent equipment corrosion and volatilization caused by dietary acidifiers during the pelleting process, disinfecting the drinking water in the waterline [6,13]. Diet-administered organic acid can be used as a more optimal performance enhancer than water-administered organic acid during the starter stage, but the final results of factors such as body weight were not different from water-administered organic acid in chicken broilers [14]. In addition, the supplement of water-administered organic acid via a water line was more convenient than dietary supplementation.

Plant essential oils such as cinnamon and eucalyptus oils exhibited wide antioxidant and anti-inflammatory activities against inflammation induced by bacterial lipopolysaccharide, a Gram-negative bacterial constituent [15]. Essential oils with different active ingredients such as cinnamaldehyde and turmeric could stimulate intestinal enzyme secretion and improve nutrient absorption [9]. The combined use of organic acids and essential oils has garnered great interest because of the synergistic effects of these additives on the intestinal microbiota and intestinal health, especially on the control of pH-sensitive foodborne pathogenic bacteria [16]. Different combinations of essential oils and organic acids exhibited different functions. A combination of cinnamaldehyde and calcium formate was effective in reducing *Salmonella* and *Clostridium* counts in ileal and caecal content, and increasing the antibody titer of the Newcastle disease vaccine; however, it did not affect growth and the feed conversion ratio [17]. Thymol-benzoic acid decreased the bacterial counts in liver, spleen, and cecum content more effectively than *Salmonella Enteritidis* treatment and surpassed the bacteriostatic effect of the cinnamaldehyde–caproic acid complex [18]. A blend of formic acid, benzoic acid, and essential oils positively affected growth performance and the gut microbiota, enhanced intestinal barrier integrity, and alleviated oxidative stress and inflammation under LPS stimulation [5]. Most studies have focused on the effects of organic acids or essential oils on broiler growth. However, the relationship between drinking water quality and the status of broiler growth or health has not been given sufficient attention.

Thus, we hypothesized that acidifiers supplemented in drinking water improved water quality and further helped to maintain intestinal function and immune status, thereby positively affecting growth. In this study, the effects of different organic acids supplemented in water, in combination with essential oils, on drinking water quality and the growth performance, intestinal digestive capacity, and immune status of broiler chickens were evaluated. Our research provides insights into the mechanism of function of acidifiers from the perspective of drinking water.

## 2. Materials and Methods

### 2.1. Bird, Diets, Drinking Water, and Experimental Design

A total of 480 1-day-old Arbore Acres broilers were collected (50% male and 50% female) and were randomly assigned to four treatments (6 replicates and 20 birds per replicate). All broilers were fed the same basal diet. The four treatment groups were the control group (Ctr), drinking water supplemented with 1.05% acidifier I (ACI), 1.05% acidifier I and 0.01% essential oil (ACI+EO), and 1.55% acidifier II (ACII). The mixed organic acid product ACI was purchased from CID LINE Co. Ltd. (Beijing, China), and primarily contains formic acid (35%), propionic acid (35%), lactate (5%), and citric acid monohydrate (1%). The mixed organic acid product ACII was purchased from Nutreco N.V. (Beijing China), and consists of formic acid (32%), ammonium formate (20%), acetic acid (7%), and copper ions (0.48%). The primary components of the essential oils were eucalyptus essential oil, mint essential oil, and cinnamaldehyde.

Acidifiers in liquid were provided every other day during the whole experimental period (0–42 days). The acidifiers were added for at least 8 h on the day of supplementation. Namely, the proportions of ACI and ACII added via the dosing pump were 1.05% and 1.55% respectively (the exact dosage was confirmed through a pretest). Essential oil was only added on days 4–10 and days 19–25 at the dose of 0.01% through the waterline, excluding day 7 and day 22 for vaccination.

The room temperature was maintained at 34–35 °C for the first day and decreased until the temperature was 21–22 °C. The basal diet was formulated to meet or exceed the nutrient requirement for broilers as recommended by NY/T 33-2004 (a standard for raising chickens in China). The ingredients and nutrient composition of the basal diet for the two phases, starter (day 0 to 21) and grower (day 22 to 42), are shown in Table 1. The experimental procedures were approved and conducted under the guidelines of the Animal Health and Care Committee of Shanxi Agricultural University (Shanxi, China).

### 2.2. Growth Performance and Sample Collection

Growth performance including average body weight and average daily gain was recorded at the phases of day 0–21 and day 22–42. On day 42, one broiler chicken of approximately average weight in each replicate was selected and killed by cervical dislocation. Duodenum, jejunum, and ileum tissue samples (1 cm) in the middle segment of each part were collected and fixed in 4% (m/vol) paraformaldehyde solution for histological examination. The contents of the crop, proventriculus, gizzard, duodenum, jejunum, ileum, and cecum were, respectively, mixed and stored at −80 °C. The liver, abdominal adipose, spleen, thymus, and bursa of Fabricius were gathered for the calculation of the internal organ index (internal organ index (%) = internal organ weight/body weight × 100%). Blood was collected through the wing vein before cervical dislocation, placed at room temperature for 1 h, and the serum was isolated by centrifugation at 1200× *g* for 10 min at 4 °C and stored at −20 °C until analysis.

### 2.3. Gastrointestinal pH, Digestive Enzyme Activity, and Intestinal Histomorphology

Gastrointestinal pH value was determined using a pH meter (Testo SE & Co. KGaA, Titisee-Neustadt, Germany) after the gastrointestinal contents were diluted in a ratio of 1:4 (m/vol) with ultrapure water and vibrated vigorously. The enzyme activity of trypsin (catalog A 080-2) and lipase (catalog number A054-1) in the small intestine was analyzed using commercial kits (Nanjing Jiancheng Bioengineering Institute, Nanjing, China), and all the procedures were conducted according to the kit manufacturer’s instructions. Formalin-fixed intestinal samples were prepared using paraffin embedding techniques. Consecutive sections (5 μm) were stained using hematoxylin and eosin for histomorphological observation. The villus height (from the tip of the villus to the crypt opening) and the crypt depth (from the base of the crypt to the crypt opening) were measured from 10 randomly selected villi and associated crypts.

### 2.4. Serum Lipid Metabolites and Immune Function

Serum lipid metabolites, including total cholesterol, total triglyceride, HDL, LDL, and free fatty acids, were determined using an automatic biochemical analyzer (Mindray BS-180, Shenzhen Mindray Bio-Medical Electronics Co., Ltd., Shenzhen, China) via the spectrophotometry method. Immunoglobulins, including IgA, IgG, and IgM, and cytokines IL-6 and IL10, were detected in the serum using commercial ELISA kits (Shanghai Enzyme-linked Biotechnology Co., Ltd., Shanghai, China). All the procedures were conducted according to the kit manufacturer’s instructions.

### 2.5. Microbial Determination

Water samples were collected at the end of the waterline on the last day of weeks 1, 2, and 5 for the determination of *Escherichia* and mold using the microbial slide counting method. Feces samples were gathered on day 42 and the numbers of *E. coli*., *Salmonella*, and *Lactobacillus* were examined using selective culture media such as MacConkey agar medium, Xylose–Lysine–Desoxycholate Agar Medium (XLD agar medium), and De Man–Rogosa–Sharp agar medium (MRS agar medium), respectively (Beijing AOBOX Biotechnology, Beijing, China). Briefly, feces were mixed with sterile water, and 0.1 mL of the homogenate was used to spread the plate. Coated plates were cultured overnight and counted.

### 2.6. Statistical Analysis

Statistical analysis between treatment levels was evaluated by one-way analysis of variance (ANOVA) and the significance among the groups was identified using the Tukey test for multiple comparisons. Statistical significance was performed with SPSS 26.0 (IBM, Armonk, NY, USA), and *p* <0.05 was considered to indicate a statistically significant difference.

## 3. Results

### 3.1. Growth Performance and Internal Organ Weights

As shown in Table 2, compared with the control group, only ACI treatment significantly decreased body weight on day 28 and average daily gain from day 0–28. Body weight on day 28 and average daily gain from day 0–28 were not significantly influenced by ACI+EO and ACII treatments compared with the control group. Compared with the control group and ACI treatment, both ACI+EO and ACII treatments significantly increased the final body weight on day 42. However, ACI+EO and ACII treatments significantly increased the average daily gain from day 29–42 and day 0–42 compared with the control group. Compared with ACI treatment, ACI+EO significantly increased the average daily gain throughout the experimental period and ACII treatments significantly increased the average daily gain from day 0–28 and day 0–42.

In Table 3, the relative weight of the liver, abdominal adipose, thymus, and bursa of Fabricius were not influenced by any treatment. The relative weight of the spleen was significantly increased by ACI+EO and ACII treatments compared with the control group.

### 3.2. Serum Immune Indicators and Lipid Metabolites

The serum immune indicators are shown in Table 4. Compared with the control group, both ACI and ACI+EO treatments reduced the level of serum IL-6, and the ACI+EO treatment also decreased the level of IL-10. No significant difference between ACII and the control group was detected regarding levels of serum IL-6 and IL-10. Compared to the control group, ACI treatment significantly increased serum IgA but decreased the levels of serum IgG and IgM, while ACII treatments significantly increased the level of serum IgA and IgM compared to the control group, albeit to a lesser extent than ACI+EO treatment.

In Table 5, compared with the control group, ACI+EO treatment significantly decreased the levels of serum total cholesterol and LDL. Similarly, ACII treatment also significantly decreased the levels of serum total cholesterol, total triglyceride, and LDL but increased free fatty acid content.

### 3.3. Gastrointestinal pH Value and Digestive Enzyme Activity

As shown in Table 6, ACI+EO treatment reduced the pH value of the crop and gizzard, and lower pH values in the proventriculus and gizzard were observed with ACII treatments compared to the control group and ACI treatment. The pH of the duodenum, jejunum, ileum, and cecum was not significantly influenced by any treatment.

Trypsin activity in the duodenum, jejunum, and ileum was not significantly influenced. The activity of lipase in the jejunum was significantly enhanced by ACI, ACI+EO, and ACII treatments compared with the control group, but the activity of lipase in the duodenum and ileum was not significantly influenced by any treatment.

### 3.4. Intestinal Morphology

Intestinal morphology on day 42 is shown in Table 7. Compared with the control group, the intestinal morphology, including villus height, crypt depth, and the ratio of villus height and crypt depth, was not significantly influenced by the supplement of different acidifiers or essential oils. Both ACI and ACI+EO treatments tended to increase the intestinal length of the jejunum (*p* = 0.062) and ileum (*p* = 0.075).

### 3.5. Potential Pathogenic Microorganisms

In Figure 1a,b, supplementation with different acidifiers and essential oils all promoted a rapid decline in *Escherichia* numbers in the waterline in the second week, while only ACI and ACI+EO treatments decreased the number of molds in the waterline in the second week. ACII was not effective in the inhibition of mold growth. However similar levels of *Eschrichia* and mold in all groups were observed in the fifth week. The numbers of *E. Coli*, *Salmonella*, and *Lactobacillus* in excreta were determined (Figure 1c). ACI and ACI+EO treatments significantly decreased the number of *Escherichia* in excreta, and only ACII significantly decreased the number of *Salmonella* in excreta compared with the control group. However, ACI, ACI+EO, and ACII treatments all significantly increased *Lactobacillus* in excreta relative to the control group.

## 4. Discussion

Organic acids can help to maintain intestinal health, improving weight gain, live weight, and immunological response, when used in conjunction with excellent nutrition management. Essential oils and organic acids are considered promising alternatives to antibiotic growth promoters to improve the growth performance and gut health of chickens due to their strong antimicrobial and antioxidant effects [19].

In this study, ACI (with formic acid, propionic acid, and lactic acid as major ingredients) did not significantly promote the growth performance on days 29–42. The influence of formic acid on growth performance was not constant. Formic acid administration in drinking water decreased the body weight of broilers at 21 and 42 day of age [20,21]. Overuse of organic acids may reduce water and feed intake due to the strong taste or sub-clinical intestinal problems and lead to the depressed growth performance of broilers. ACII (with formic acid, ammonium formate, and acetic acid as major ingredients) and ACI with essential oils showed better growth performance than the control group during the finisher period. Similar results were found in another study, which showed that acidification of drinking water using propionic acid, ammonium propionate, formic acid, and ammonium formate as active ingredients effectively improved the body weight, average daily gain, and feed conversion ratio [6]. The mixture of organic acids (benzoic acid, formic acid, and lactic acid), which is similar to ACI, did not significantly influence growth performance during the starter, grower, and finisher phases, or during the whole period, while a combination of organic acids and essential oils (cinnamaldehyde, carvacrol, thymol, and eugenol) significantly decreased the feed conversion ratio [19]. Essential oil extracts from cinnamon, oregano, and rosemary, with cinnamaldehyde as one of the active compounds, improved the apparent ileal digestibility of nutrients and ether extract digestibility [22]. Increased body weight gain after the use of essential oils and ACII could be closely related to their antibacterial effects and the modulation of pH, as well as the digestive capacity of the gastrointestinal tract.

The liver is an important organ for nutrient metabolism, a sensitive indicator of toxicity, and the major lipogenic tissue in poultry [23,24]. In our study, the relative weights of the liver and abdominal adipose were not significantly influenced by supplementation with mixed acidifiers and essential oils. Likewise, the unchanged relative weight of the liver after dietary mixed organic acid supplements on days 21 and 42 in broilers was also observed [25]. The spleen is involved in the humoral and cellular immune response through modulation of lymphocytes [26]. The increased relative weight of internal organs indicated beneficial immunological advances, and supplementation of formic acid improved the relative weights of lymphoid organs such as the spleen, bursa of Fabricius, and thymus [27]. In this study, the increased relative weight of the spleen caused by ACI+EO and ACII treatments suggested the enhanced immune function of broilers, but the relative weight of the thymus and bursa of Fabricius was not significantly influenced. Administration of acidifiers significantly negated the increasing trend of ileal IL-6 expression caused by *E. coli* [28]. Similarly, in this study, ACI alone or together with essential oils significantly decreased the level of IL-6. Mixed organic acid supplementation increased the serum IgA level in broilers [3]. Increased serum IgA might be related to the potential boost of B and T lymphocytes and organic acid could enhance the immune status and lymphocyte responses in the gut [29]. In this study, ACI with essential oils was the most effective treatment, inhibiting IL-6 levels and increasing levels of IgA and IgM, which contribute to the enhancement of immune status.

ACII significantly decreased total cholesterol, total triglyceride, and low-density lipoproteins (LDLs) and increased free fatty acid levels in the serum. Fatty acids derived from the diet or the liver are transported to adipose tissue via LDL or chylomicrons [30]. Decreased total cholesterol and LDL were also detected by ACI with essential oil treatment. Decreased LDL may contribute to lower levels of total cholesterol and triglyceride, which indicate a normal lipid metabolism status in the liver. The digestion and absorption of nutrients, including lipids, were primarily determined by intestinal function. The intestinal digestive capacity was further examined.

The digestive tract of poultry is relatively short and the pH of the small intestine is vulnerable to external environmental factors [3]. In this study, supplementation with liquid acidifiers and essential oils in drinking water only reduced the pH of the anterior segment of the digestive tract, including the crop, proventriculus, and gizzard, while the pH levels of the small intestine and cecum were not changed. The effect of mixed organic acids on small intestinal pH was mainly centered in the anterior segment or had no significant effects on the pH value of the duodenum, jejunum, and ileum of broilers [3,31]. The pH of the jejunum was not affected by dietary formic acid [21]. An unprotected organic acid mixture was usually considered to be active in the foregut and neutralized by bile, which only affects the upper intestine by reducing stomach pH [32]. On the other hand, the high buffering capacity of feed or insufficient dosage may contribute to the lack of effect the acidifiers had on the pH of the digesta from the hindgut [33]. Increased acid concentrations in the crop could contribute to controlling infections caused by horizontal transmission and a warm moist crop environment, further enhancing the antibacterial activity of short-chain fatty acids [34].

Broiler performance can also be affected by the activities of endogenous intestinal enzymes [35]. The activity of the digestive enzymes trypsin and lipase in the small intestine was measured and only jejunal lipase activity was significantly increased by acidifier supplementation. However, the morphology of the small intestine was not influenced by mixed liquid acidifiers and essential oils in this study. A blend of short- and medium-chain fatty acids alleviated the decreased villus length caused by *S. Typhimurium* [36], which indicated that organic acids may benefit the improvement of small intestinal morphology only under challenge.

The water quality of the water system plays an important role in the health and growth performance of broiler chickens [37]. *E. coli* and *Salmonella* are frequently found in drinking water and both microorganisms are capable of forming biofilms in poultry environments, which makes waterlines potential sources of pathogens in chickens [12,37]. Mold growth was associated with the production of mycotoxin and organic acids were highly effective against mold and bacteria [10]. Organic acids have long been applied in poultry feeds to combat bacterial and fungal pollution [32]. Formic acid showed higher antibacterial activity against *E. coli* and *Salmonella* spp., followed by propionic acid [7]. Propionic acid exhibited a stronger inhibitory effect against fungal growth than acetic acid, probably through intracellular acidification, toxic anion accumulation, and membrane disruption [38]. In addition, a lower level of mold in the waterline was observed in the ACI and ACI+EO groups than in the ACII group. Essential oils (cinnamaldehyde, thymol, and eucalyptus essential oil) were effective in inhibiting the growth of *Clostridium perfringens* in vitro and decreased the intestinal lesion score [39]. The antibacterial effects of organic acids and essential oils were enhanced by each other. As essential oils usually have great hydrophobicity and increase the permeability of the bacterial membrane, more organic acids under the undissociated form were allowed to penetrate the bacterial cytoplasm, causing the death of pH-sensitive bacteria such as *E. coli*, *Salmonella*, and *C. perfringens* [16]. On the other hand, organic acids can also increase the concentration of H+, which allows essential oils to exist in a molecular form that can freely enter the bacterial cell and exert its antimicrobial activity [40]. Growth inhibition of *Escherichia* and mold was observed after supplementation with acidifiers and essential oils during the starter phase and the growth phase in our study. Controlling the introduction and transmission of pathogens was the goal of preharvest pathogen reduction, and drinking water treatment and litter management are common *Salmonella* intervention methods [41]. Perhaps the supply of acidified water used in the water line was harmful or even lethal to bacteria and cleaned the drinking water. However, a discontinuous supply of acidified water was adapted by the host and induced an adaptive tolerance response in intestinal bacteria, and the modulation of intestinal microbiota might produce a positive effect on the health of broilers.

Organic acid consisting of formic acid, acetic acid, and ammonium formate in water decreased the amounts of *Salmonella* in cecal chyme [11]. Acetic acid was one of the components of ACII. Many butyrate-producing microbiota were identified as net utilizers of acetate, and butyrate provided fuel for epithelial cells and contributed to intestinal health [42]. In addition, increased butyrate was correlated with *lactic acid bacteria* [43]. In this study, both ACI+EO and ACI increased *Lactobacillus* in the excreta. Organic acids have a particular proclivity to kill Gram-negative bacteria, while some acid-tolerant Gram-positive beneficial bacteria such as *Lactobacillus* and *Bifidobacterium* can survive at lower pH levels [44]. Thus, further study is needed to reveal the changes in gut microbes.

## 5. Conclusions

In conclusion, a combination of ACI and EO enhanced the immune status by increasing serum antibodies and anti-inflammation function, while ACII was effective in reducing *Salmonella* and promoting the growth of *lactobacillus*, which contributed to intestinal health. In addition, the improved water quality may contribute to the growth-promoting effects of ACI+EO and ACII.

## Figures and Tables

**Figure 1 animals-14-02160-f001:**
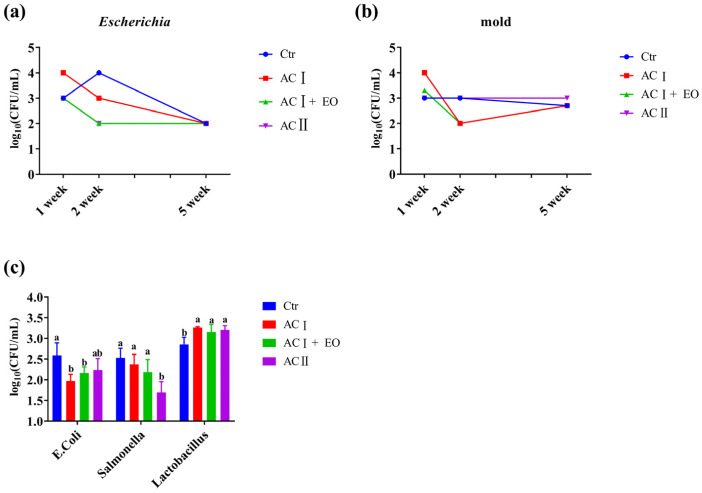
Effects of mixed liquid acidifier and essential oil on the growth of microorganisms. (**a**) Growth of *E. coli* in the waterline on weeks 1, 2, and 5; (**b**) Growth of mold in the waterline on weeks 1, 2, and 5; (**c**) The number of *E. coli*, *Salmonella*, and *Lactobacillus* in excreta. Note: In (**a**,**b**), the growth of microorganisms in the waterline was numerically described; in (**c**), means within a row lacking a common superscript differ (*p* < 0.05).

**Table 1 animals-14-02160-t001:** Ingredients and nutrient compositions of the basal diets.

Ingredients, %	Starter Diet	Finisher Diet
Corn	51.169	56.559
Soybean meal	35.835	31.400
Wheat flour	5.000	3.219
Soybean oil	4.000	5.500
Limestone	1.348	0.581
Dicalcium phosphate	1.248	1.533
NaCl	0.300	0.300
L-Lysine (78%)	0.356	0.170
DL-Methionine (98%)	0.215	0.208
Choline chloride (50%)	0.150	0.150
Mineral premix ^1^	0.200	0.200
Enzyme blend ^2^	0.030	0.030
Vitamin premix ^3^	0.030	0.030
Phytase (5000 FTU/kg)	0.020	0.020
Additive of Exp ^4^	0.100	0.100
Nutrient composition ^4^, %		
ME (mc/kg)	3.00	3.15
CP, %	21.50	19.50
Total Lysine, %	1.41	1.16
Total Methionine, %	0.53	0.50
Total Methionine + Cysteine, %	0.85	0.80
Calcium, %	1.00	0.79
Available phosphorus, %	0.35	0.39

^1^ The mineral premix provided per kg diet: CuSO_4_·5H_2_O, 8 mg; FeSO_4_, 80 mg; MnSO_4_·H_2_O, 100 mg; Na_2_SeO_3_, 0.15 mg; KI, 0.35 mg. ^2^ Enzymes provided per kg diet: xylanase ≥ 2250 viscosity unit; β-dextranase ≥ 52 AGL U/g. ^3^ The vitamin premix provided per kg diet: vitamin A, 9500 IU; vitamin D3, 62.5 μg; vitamin E 300 IU; vitamin K3, 2.65 mg; vitamin B6, 6 mg; vitamin B12, 0.025 mg; biotin, 0.0325 mg; folic acid, 1.25 mg; pantothenic acid, 12 mg; nicotinic acid 50 mg. ^4^ Nutrient composition of basal diets was based on calculations.

**Table 2 animals-14-02160-t002:** Effects of mixed liquid acidifier and essential oil on body weight.

Item	Treatment ^1^				SEM	*p*-Value
	Ctr	ACI	ACI+EO	ACII		
Average Body Weight						
Day 0/g	43.0	42.3	43.3	43.6	0.020	0.128
Day 28/kg	1.56 ^a^	1.50 ^b^	1.58 ^a^	1.57 ^a^	0.008	0.001
Day 42/kg	2.74 ^b^	2.71 ^b^	2.83 ^a^	2.79 ^a^	0.012	<0.001
Average Daily Gain						
Day 0–28/g	54.13 ^a^	52.18 ^b^	54.89 ^a^	54.40 ^a^	0.279	0.001
Day 29–42/g	84.25 ^c^	86.09 ^bc^	89.35 ^a^	87.72 ^ab^	0.622	0.014
Day 0–42/g	64.17 ^b^	63.48 ^b^	66.37 ^a^	65.51 ^a^	0.295	<0.001

^1^ Ctr: basal diet without supplementations in drinking water; ACI: liquid acidifier I supplemented in drinking water; ACI+EO: liquid acidifier I and essential oil supplemented in drinking water; ACII: liquid acidifier II supplemented in drinking water. Different letters on the same line are different according to Tukey’s test.

**Table 3 animals-14-02160-t003:** Effects of mixed liquid acidifier and essential oil on internal organ index on day 42.

Item	Treatment ^1^				SEM	*p*-Value
	Ctr	ACI	ACI+EO	ACII		
Liver %	2.19	2.29	2.22	2.19	0.026	0.896
Abdominal adipose %	1.45	1.64	1.41	1.61	0.134	0.817
Spleen %	0.08 ^b^	0.11 ^ab^	0.14 ^a^	0.13 ^a^	0.007	0.003
Thymus %	0.19	0.24	0.22	0.20	0.016	0.528
Bursa of Fabricius %	0.04	0.05	0.04	0.04	0.004	0.670

^1^ Ctr: basal diet without supplementations in drinking water; ACI: liquid acidifier I supplemented in drinking water; ACI+EO: liquid acidifier I and essential oil supplemented in drinking water; ACII: liquid acidifier II supplemented in drinking water. Different letters on the same line are different according to Tukey’s test.

**Table 4 animals-14-02160-t004:** Effects of mixed liquid acidifier and essential oil on serum immune status on day 42.

Item	Treatment ^1^				SEM	*p*-Value
	Ctr	ACI	ACI+EO	ACII		
IL-6 (ng/L)	50 ^a^	40 ^c^	45 ^b^	48 ^a^	0.7	<0.001
IL-10 (ng/L)	60 ^a^	62 ^a^	54 ^b^	61 ^a^	0.6	<0.001
IgA (ng/L)	5908 ^c^	7005 ^a^	7136 ^a^	6639 ^b^	93.3	<0.001
IgG (ng/L)	86 ^ab^	80 ^c^	87 ^a^	83 ^bc^	0.7	<0.001
IgM (ng/L)	4439 ^c^	4045 ^d^	5061 ^a^	4740 ^b^	72.8	<0.001

^1^ Ctr: basal diet without supplementations in drinking water; ACI: liquid acidifier I supplemented in drinking water; ACI+EO: liquid acidifier I and essential oil supplemented in drinking water; ACII: liquid acidifier II supplemented in drinking water. Different letters on the same line are different according to Tukey’s test.

**Table 5 animals-14-02160-t005:** Effects of mixed liquid acidifier and essential oil on serum lipid metabolites on day 42.

Item	Treatment ^1^				SEM	*p*-Value
	Ctr	ACI	ACI+EO	ACII		
Total cholesterol (mmol/L)	3.18 ^a^	2.90 ^ab^	2.63 ^b^	2.59 ^b^	0.066	0.002
Total triglyceride (mmol/L)	1.17 ^a^	0.92 ^ab^	0.73 ^ab^	0.57 ^b^	0.071	0.012
HDL (mmol/L)	2.01	1.89	1.75	1.87	0.053	0.407
LDL (mmol/L)	0.66 ^a^	0.57 ^ab^	0.48 ^b^	0.49 ^b^	0.026	0.033
Free fatty acid (μmol/L)	320.16 ^b^	338.91 ^b^	386.42 ^b^	549.59 ^a^	21.878	<0.001

^1^ Ctr: basal diet without supplementations in drinking water; ACI: liquid acidifier I supplemented in drinking water; ACI+EO: liquid acidifier I and essential oil supplemented in drinking water; ACII: liquid acidifier II supplemented in drinking water. Different letters on the same line are different according to Tukey’s test.

**Table 6 animals-14-02160-t006:** Effects of mixed liquid acidifier and essential oil on internal organ index on day 42.

Item		Treatment ^1^				SEM	*p*-Value
		Ctr	ACI	ACI+EO	ACII		
pH value	Crop	4.08 ^a^	4.16 ^a^	3.40 ^b^	3.75 ^ab^	0.544	0.003
	Proventriculus	3.63 ^a^	3.61 ^a^	3.69 ^a^	3.19 ^b^	0.057	0.003
	Gizzard	3.61 ^a^	3.55 ^a^	3.10 ^b^	3.04 ^b^	0.060	<0.001
	Duodenum	5.01	5.07	4.79	5.04	0.360	0.280
	Jejunum	5.41	5.17	5.11	5.45	0.439	0.215
	Ileum	5.79	5.50	5.74	5.91	0.372	0.082
	Cecum	6.06	6.07	6.21	6.17	0.259	0.542
Trypsin activity (U/mgprot)	Duodenum	2880.23	2248.96	3409.75	4200.14	378.49	0.326
	Jejunum	5454.55	7480.88	9372.99	6839.91	663.21	0.214
	Ileum	11,522.79	8425.48	8925.59	7214.97	727.15	0.199
Lipase activity (U/mgprot)	Duodenum	136.11	66.46	69.15	67.86	16.26	0.364
	Jejunum	157.04 ^b^	486.55 ^a^	429.94 ^a^	474.88 ^a^	48.51	0.040
	Ileum	743.34	534.27	532.84	417.94	60.26	0.296

^1^ Ctr: basal diet without supplementations in drinking water; ACI: liquid acidifier I supplemented in drinking water; ACI+EO: liquid acidifier I and essential oil supplemented in drinking water; ACII: liquid acidifier II supplemented in drinking water. Different letters on the same line are different according to Tukey’s test.

**Table 7 animals-14-02160-t007:** Effects of mixed liquid acidifier and essential oil on internal organ index on day 42.

Item		Treatment ^1^				SEM	*p*-Value
		Ctr	ACI	ACI+EO	ACII		
Intestinal length/cm	Duodenum	26.2	28.0	27.3	25.2	0.55	0.308
	Jejunum	62.0	71.0	66.0	59.1	1.67	0.062
	Ileum	49.7	58.5	53.8	52.2	1.24	0.075
Villus height/μm	Duodenum	1804	1822	1762	1759	33.7	0.895
	Jejunum	1237	1374	1425	1378	51.1	0.615
	Ileum	836	733	1009	983	45.3	0.101
Crypt depth/μm	Duodenum	255	251	244	208	8.1	0.165
	Jejunum	212	200	173	205	6.5	0.180
	Ileum	139	154	153	158	5.2	0.638
V/C ^2^	Duodenum	7.57	7.38	7.74	8.94	0.261	0.139
	Jejunum	7.20	7.08	7.20	6.91	0.207	0.959
	Ileum	6.38	5.14	6.23	5.99	0.209	0.150

^1^ Ctr: basal diet without supplementations in drinking water; ACI: liquid acidifier I supplemented in drinking water; ACI+EO: liquid acidifier I and essential oil supplemented in drinking water; ACII: liquid acidifier II supplemented in drinking water; ^2^ V/C: (villus height)/(crypt depth), μm/μm.

## Data Availability

Data are contained within the article.

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
