# Peer review of "Effects of Mixed Organic Acids and Essential Oils in Drinking Water on Growth Performance, Intestinal Digestive Capacity, and Immune Status in Broiler Chickens"

_animals, 2024, doi:10.3390/ani14152160_

Round 1

Reviewer 1 Report

Comments and Suggestions for Authors

The article provides useful information about using combination of organic acids and essential oils as green alternatives to antibiotics, improving growth performance and gut health in broiler chickens. Organic acids exhibit antimicrobial properties and enhance nutrient absorption. Additionally, the mixed organic acids and essential oils shows synergistic effects on intestinal microbiota and health. However, there are some critical points that need to be addressed:

1.                  According to the authors' guidelines, the abstract section should have a maximum of 200 words.

2.                  In the Introduction, the novelty or 'gaps in knowledge' of the current work have not been adequately mentioned. Similar articles can be found in the literature. The authors are encouraged to explicitly state the novelty of their work in the introduction section.

3.                  Please add the hypothesis of the study.

4.                  Materials and Methods:

-L117 Add information about sex of birds.

- L118 consider changing "…four treatments (6 replicates and 20 birds per replicate" at L.28 and L.118.

-Table 1- How did it look analyze nutrient composition in diets (CP, Ca, P, Lys, Met)? The metabolizable energy was calculate or analysed?  There is a lack in methods.

- L157-160- Did you take the blood post mortem? You need add this information.

5.Results:

- Table 2- Growth Performance: Why didn't you measure feed intake and calculate the feed conversion ratio? These are two important measurements. You should add this, especially since some feed additives can enhance the appetite of birds.

-The entire Results section requires rewriting. The description of the results is inadequate compared to what is presented in the tables, for example: L203 and 204. Descriptions should be more concise and clear.

- There are two tables with number 6.

6. Discussion:

The Discussion section requires improvement. Currently, it mainly repeats results and compares them with existing literature, with little effort made to explain the mechanisms of actions.

Author Response

Author's Reply to the Review Report

Comments 1: According to the authors' guidelines, the abstract section should have a maximum of 200 words.

Response 1:

Revised as suggestion.

Comments 2:  In the Introduction, the novelty or 'gaps in knowledge' of the current work have not been adequately mentioned. Similar articles can be found in the literature. The authors are encouraged to explicitly state the novelty of their work in the introduction section.

Response 2:

Revised as suggestion.

Comments 3: Please add the hypothesis of the study.

Response 3:

Revised as suggestion. Hypothesis was added in introduction.

Comments 4: Materials and Methods:

-L117 Add information about sex of birds.

- L118 consider changing "…four treatments (6 replicates and 20 birds per replicate" at L.28 and L.118.

-Table 1- How did it look analyze nutrient composition in diets (CP, Ca, P, Lys, Met)? The metabolizable energy was calculate or analysed?  There is a lack in methods.

- L157-160- Did you take the blood post mortem? You need add this information.

Response 4:

-Line 117 The sex of birds used in our study was added.

-Line 118 Revised as suggestion.

-Table 1 Nutrient composition in diets including CP, Ca, P, Lys, Met, ME was based on calculation. Revised on the note of table 1.

-Line 157-160 Blood was collected before broilers were killed for cervical dislocation.

Comments 5:

Results:

- Table 2- Growth Performance: Why didn't you measure feed intake and calculate the feed conversion ratio? These are two important measurements. You should add this, especially since some feed additives can enhance the appetite of birds.

-The entire Results section requires rewriting. The description of the results is inadequate compared to what is presented in the tables, for example: L203 and 204. Descriptions should be more concise and clear.

- There are two tables with number 6.

Response 5:

-Table 2 Due to the limitations of facility, in the automatic chicken feeder feeding system we can only measure the total feed intake of each treatment arranged alongside the waterline. Thus, there was no statistical analysis between treatments, but only a simple description of feed intake was recorded. (the feed intake and FCR was shown below)

Item

Treatment1

Ctr

ACâ… 

ACâ… +EO

ACâ…¡

SEM2

p values

Average Body Weight /kg

Day 28

1.56a

1.50b

1.58a

1.57a

0.008

0.001

Day 42

2.74b

2.71b

2.83a

2.79a

0.012

<0.001

Average Daily Gain /g

Day 0-28

54.13a

52.18b

54.89a

54.40a

0.279

0.001

Day 29-42

84.25c

86.09bc

89.35a

87.72ab

0.622

0.014

Day 0-42

64.17b

63.48b

66.37a

65.51a

0.295

<0.001

Total feed intake/ kg

Day 0-42

4.33

4.30

4.30

4.30

-

-

FCR

Day 0-42

1.64

1.64

1.56

1.58

-

-

Survival rate/ %

Day 0-42

95.02

95.77

96.02

96.00

-

-

EPI

Day 0-42

378.86

375.53

415.44

402.04

-

-

-Results section were carefully checked and rewritten including L203 and 204.

-table number was revised.

Comments 6: The Discussion section requires improvement. Currently, it mainly repeats results and compares them with existing literature, with little effort made to explain the mechanisms of actions.

Response: Discussion section was carefully checked and rewritten

Reviewer 2 Report

Comments and Suggestions for Authors

Please find comments and suggestions for authors in the attached file

Author Response

Author's Reply to the Review Report

Comments 1:

Line 17: are the reduced potential corrosion of processing equipment and lose during granulation the only positive effect related to acidifier added in drinking water?

Response 1:  

Besides the positive effects mentioned above, acidifiers in liquid were more easily to supplemented through water line according the requirement of production and helped to keep the water quality. The improved water quality was determined in this text.

Comments 2:

Line 21: specify the EO considered in this study (eucalyptus essential oil, mint essential oil and cinnamaldehyde)

Response 2:  

Revised as suggestion.

Comments 3:

Line 30: please better specify the treatment groups diet composition

Response 3:  

Revised as suggestion in the text. Broilers were fed the same basal diet.

Comments 4:

Line 31: please explain the reason why EO was only added on day 4-10 and day 19-25

Response 4:  

Firstly, add essential oil at intervals to reduce the production costs. Secondly, as intestinal microbiota achieved stability at 21 days of age (Zhou et al., 2021), and diet was changed from stater phase to finisher phase on day 21. Similar to the intermittent acidic water supply to induce defense mechanism in bacteria to defend the cell from acid (Hamid et al., 2018). Discontinuous supply of essential oil may be more effective than constant supply. Thus, early addition of essential on 4-10 may be effective to modulate intestinal microbiota, and addition on 19-25 may contribute to maintaining the microbiota homeostasis.

Zhou, Q., F. Lan, X. Li, W. Yan, C. Sun, J. Li, N. Yang, and C. Wen. 2021. The Spatial and Temporal Characterization of Gut Microbiota in Broilers. Front Vet Sci 8:712226.

Hamid, H., H. Q. Shi, G. Y. Ma, Y. Fan, W. X. Li, L. H. Zhao, J. Y. Zhang, C. Ji, and Q. G. Ma. 2018. Influence of acidified drinking water on growth performance and gastrointestinal function of broilers. Poultry Sci 97(10):3601-3609.

Comments 5:

Line 33: why the authors choose to refer to European performance index if the diet is based on two periods (starter finisher)?

Response 5:  

The results of European performance index were deleted as suggestion.

Comments 6:

Line 35: do the Authors mean distal gastrointestinal tract?

Response 6:  

Yes. Revised as suggestion.

Comments 7:

Line 55: please change in Salmonella

Response 7:  

Revised as suggestion.

Comments 8:

Line 59: better explain why OA can be considered green alternative

Response 8: 

“Mixed organic acids (MOA) are favored by feed and livestock enterprises because of their advantages of improving the growth performance and intestinal health of livestock, as well as the benefits of the three-free (i.e., drug resistance free, residue-free, and pollution-free), which make the MOA the most promising green alternatives to antibiotics (Ma et al., 2021).”

Ma, J., S. Mahfuz, J. Wang, and X. Piao. 2021. Effect of Dietary Supplementation with Mixed Organic Acids on Immune Function, Antioxidative Characteristics, Digestive Enzymes Activity, and Intestinal Health in Broiler Chickens. Front Nutr 8:673316.

Comments 9:

Line 81: please add some more detail on the digestive process related to OA ingestion

Response 9:  

Revised as suggestion.

Comments 10:

Line 117: Arbore Acres, use it before the acronym AA

Response 10:  

Revised as suggestion.

Comments 11:

Line 118: 6 cages or pens?

Response 11:  

Revised as suggestion. We use cages as replicates.

Comments 12:

Line 119: Define the criteria for choosing the dose used. Why were different concentrations not included in this study?

Response 12:  

Based on the pretest, different concentrations of acidifiers 1 and 2 was used to ensure the pH of water at 3.8-4.0.

Comments 13:

Line 121-125: The mixed organic acid product purchased from Nutreco N.V., consists of formic acid, ammonium formate, acetic acid, and copper ion. Primary components of essential oil were eucalyptus essential oil, mint essential oil and cinnamaldehyde. In which proportions/concentrations?

Response 13:  

The proportions were supplemented. Revised as suggestion.

Comments 14:

Line 134: see comment Line 33. Why considering only two phases when the arbore acres guidelines indicates starter, grower and finisher phase?

Response 14:  

Firstly, the chicken farm production required a simplified feed process, and two-phase diet also met the nutritional requirement but not precise enough. Secondly, all experimental treatments were fed the same diet and the diets did not influence the results in the text.

Comments 15:

Table 1: indicate the protein level of the used soybean meal, change in soybean oil, salt is Sodium chloride? Specify if total or available values for Amino acids levels

Response 15: 

The protein level of soybean meal used was 45%, and decreased soybean oil in finisher diet was to met the energy requirement. Salt is NaCl. For nutrient composition, total amino acids values were shown in the table.

Comments 16:

Line 150: change in recorded. One chick per pen?

Response 16:

We raised 20 birds per pen and one bird was selected for sampling.

Comments 17:

Line 162: change “this” with “the” and reformulate the sentence since is not clear

Response 17:  

Revised as suggestion.

Comments 18:

Line 167 change “according to the protocol of the kits” with “according to the kit manufacturer’s instructions”

Response 18:  

Revised as suggestion.

Comments 19:

Line 175: specify the brand of the automatic biochemical analyser used

Response 19:  

Revised as suggestion.

Comments 20:

Line 178: change “according to the protocol of the kits” with “according to the kit manufacturer’s instructions”

Response 20:  

Revised as suggestion.

Comments 21:

Line180 Microbiome determination is not correct since the results reported are referred to microbial determinations.

Response 21:  

Revised as suggestion.

Comments 22:

Line 182: Escherichia

Response 22:  

Revised as suggestion.

Comments 23:

Line 190: remove “this”

Response 23:  

Revised as suggestion.

Comments 24:

Line 192: please change “Statistical significance” with “statistical analysis”

Response 24:  

Revised as suggestion.

Comments 25:

Line 197: remove “this” and reformulate the sentence since not clear

Response 25:  

Revised as suggestion.

Comments 26:

Table 2: it could be useful to evaluate also the ADFI and the FCR since acidifiers may negatively influence the palatability of water. Therefore, a lower water intake can be related to a lower feed intake and reduced growth performances as in the case of AC 1 group during 0-28 period. Please specify the starting BW of animals, otherwise consider only the 28-42 interval.

Response 26:  

Due to the limitations of facility, in the automatic chicken feeder feeding system we can only measure the total feed intake of each treatment arranged alongside the waterline. Thus, there was no statistical analysis between treatments, but only a simple description of feed intake was recorded. (the feed intake and FCR was shown below).

Starting BW of animals was not significantly different.

Revised as suggestion.

Item

Treatment1

Ctr

ACâ… 

ACâ… +EO

ACâ…¡

SEM2

p values

Average Body Weight /kg

Day 28

1.56a

1.50b

1.58a

1.57a

0.008

0.001

Day 42

2.74b

2.71b

2.83a

2.79a

0.012

<0.001

Average Daily Gain /g

Day 0-28

54.13a

52.18b

54.89a

54.40a

0.279

0.001

Day 29-42

84.25c

86.09bc

89.35a

87.72ab

0.622

0.014

Day 0-42

64.17b

63.48b

66.37a

65.51a

0.295

<0.001

Total feed intake/ kg

Day 0-42

4.33

4.30

4.30

4.30

-

-

FCR

Day 0-42

1.64

1.64

1.56

1.58

-

-

Survival rate/ %

Day 0-42

95.02

95.77

96.02

96.00

-

-

EPI

Day 0-42

378.86

375.53

415.44

402.04

-

-

Comments 27:

Line 261: “Intestinal Length and Morphology” delete and leave only intestinal morphology

Response 27:  

Revised as suggestion.

Comments 28:

Line 275: Escherichia

Response 28:  

Revised as suggestion.

Comments 29:

Discussion: The authors should discuss the controversary findings on growth performances in a more precise way, since it is not that rare to find reduced performances due to lower palatability of water when considering acidifiers administration. Line 395-399: please, discuss only shown data or delete

Response 29:  

Discussion was revised. Line 395-399 was deleted as suggestion.

Comments 30:

Conclusion: conclusion is too generic, specify what was changed by the treatments. Please enlarge the conclusions referring to the synergistic effect of EO and AC in promote the immune status and the need of using different dose of AC to reduce pathogen bacteria.

Response 30:  

Revised as suggestion.

Comments 31:

References: page number is missing for some references

Response 31:  

Revised as suggestion.

Reviewer 3 Report

Comments and Suggestions for Authors

This study evaluates the use of mixed organic acids and essential oils as feed additives in poultry. The effects were investigated on growth performance, intestinal digestive capacity and immune status in broilers. Some points should be addressed before the manuscript can be considered for publication.

General comments 

The manuscript is well-structured. However, a major spell and style checker are required. The research design is appropriate with the proposed objectives, the results are clearly described but discussion can be improved. 

Specific comments are discussed below:

  • The abstract should be a total of about 200 words maximum.

  • Introduction: Check that the names of the bacteria are spelled correctly. There are also some typos like on line 66 (Pka).

  • Materials and methods:

  • Why did not you include a group with acidifier II and essential oils?

  • Table 1: Revise its footnote.

  • There are many typos throughout the text (e.g. The use of this instead of the; lack of italics in bacteria name, etc). Please correct it.

  • Line 180. Did you spread a single dilution of the samples?

  • Statistical analysis: Did all the variables meet the conditions to apply ANOVA? Did you use a non-parametric test?

  • Results:

  • There are many typos throughout the text (e.g. The use of this; lack of italics in bacteria names, etc). Please correct it.

  • Please, improve table formatting.

  • Figure 1. If possible, improve resolution and add error bars (a and b).

  • Discussion: the authors should include more information in this section and improve the discussion of results.  

  • References: check for correct formatting. Please also check all references are present in both the text and the list.

Comments on the Quality of English Language

Moderate editing of English language and style are required.

Author Response

Author's Reply to the Review Report

Comments 1:

The abstract should be a total of about 200 words maximum.

Response 1:  

Revised as suggestion.

Comments 2:

Introduction: Check that the names of the bacteria are spelled correctly. There are also some typos like on line 66 (Pka).

Response 2:

Revised as suggestion.

Comments 3:     

Materials and methods:

Why did not you include a group with acidifier II and essential oils?

Response 3:

In this study ACâ…  was a new product to be tested and its combination function of essential oil on broilers was determined. We used ACâ…¡, a widely used commercial acidifier as positive control to evaluate the effects of ACâ…  and its combination of essential oil.

Comments 4:

Table 1: Revise its footnote.

Response 4:

Revised as suggestion.

Comments 5:

There are many typos throughout the text (e.g. The use of this instead of the; lack of italics in bacteria name, etc). Please correct it.

Response 5:

we carefully check through the text and revised as suggestion.

Comments 6:

Line 180. Did you spread a single dilution of the samples?

Response 6:

Both the stock and diluent of the samples were determined.

Comments 7:

Statistical analysis: Did all the variables meet the conditions to apply ANOVA? Did you use a non-parametric test?

Response 7:

All the experimental data were tested for normality and homogeneity of variance before applying ANOVA, and meet the condition to apply ANOVA.

Comments 8:

Results:

There are many typos throughout the text (e.g. The use of this; lack of italics in bacteria names, etc). Please correct it.

Response 8: We carefully check through the whole text and revised as suggestion.

Comments 9:

Please, improve table formatting.

Response 9:

Revised as suggestion.

Comments 10:

Figure 1. If possible, improve resolution and add error bars (a and b).

Response 10:

Revised as suggestion. Resolution was improved. For fig 1 a, b one sample was collected in the waterline for each treatment on 1, 2, 5 week for the determination of growth of E. coli.. Thus, it is a simple description and no statistic analysis can be made.

Comments 11:

Discussion:

the authors should include more information in this section and improve the discussion of results. 

Response 11:

Revised as suggestion.

Comments 12:

References:

check for correct formatting. Please also check all references are present in both the text and the list.

Response 12:

Revised as suggestion.

Round 2

Reviewer 1 Report

Comments and Suggestions for Authors

Accept in present form

Reviewer 3 Report

Comments and Suggestions for Authors

I appreciate the modifications that you made.